# Body Processing in Children and Adolescents with Traumatic Brain Injury: An Exploratory Study

**DOI:** 10.3390/brainsci12080962

**Published:** 2022-07-22

**Authors:** Claudia Corti, Niccolò Butti, Alessandra Bardoni, Sandra Strazzer, Cosimo Urgesi

**Affiliations:** 1Scientific Institute, IRCCS E. Medea, Via Don Luigi Monza 20, 23842 Bosisio Parini, Italy; niccolo.butti@lanostrafamiglia.it (N.B.); alessandra.bardoni@lanostrafamiglia.it (A.B.); sandra.strazzer@lanostrafamiglia.it (S.S.); 2PhD Program in Neural and Cognitive Sciences, Department of Life Sciences, University of Trieste, 34127 Trieste, Italy; 3Scientific Institute, IRCCS E. Medea, 33078 San Vito al Tagliamento, Italy; cosimo.urgesi@lanostrafamiglia.it; 4Laboratory of Cognitive Neuroscience, Department of Languages and Literatures, Communication, Education and Society, University of Udine, 33100 Udine, Italy

**Keywords:** body-perception, motor-imagery, traumatic-brain-injury, brain-damage, children, adolescents, rehabilitation

## Abstract

Dysfunctions in body processing have been documented in adults with brain damage, while limited information is available for children. This study aimed to investigate body processing in children and adolescents with traumatic brain injury (TBI) (N = 33), compared to peers with typical development. Two well-known computerized body-representation paradigms, namely Visual Body Recognition and Visuo-spatial Imagery, were administered. Through the first paradigm, the body inversion and composite illusion effects were tested with a matching to sample task as measures of configural and holistic processing of others’ bodies, respectively. The second paradigm investigated with a laterality judgement task the ability to perform first-person and object-based mental spatial transformations of own body and external objects, respectively. Body stimuli did not convey any emotional contents or symbolic meanings. Patients with TBI had difficulties with mental transformations of both body and object stimuli, displaying deficits in motor and visual imagery abilities, not limited to body processing. Therefore, cognitive rehabilitation of body processing in TBI might benefit from the inclusion of both general training on visuo-spatial abilities and specific exercises aimed at boosting visual body perception and motor imagery.

## 1. Introduction

Body processing is a basic human skill that is essential for social functioning in both children and adults [1,2], as it allows both the perception of others [3] and the representation of the corporeal self [4,5,6]. This ability relies on both visual and sensorimotor information [7,8,9,10]. Efficient visual body processing allows the identification of the morphology, structure, and boundaries of one’s own and others’ body parts [11,12,13,14]. Previous studies have shown that, similar to face perception [15], body stimuli are processed through configural and holistic visual perceptual strategies, which are more efficient than detail-based processing adopted for other objects [16]. Configural processing of bodies has been inferred by the disproportionate impairment in discriminating inverted as compared to upright bodies (body inversion effect; [17,18,19,20], to an extent that doesn’t occur with other objects. Similarly, holistic processing of bodies has been documented by the tendency to report that two identical top half bodies are different when integrated with two different bottom halves (*composite illusion effect* [21]). In this respect, visual body perception engages specific areas in the occipito-temporal cortex that differ from those neural systems responsible for the representation of objects or faces [22,23,24,25] and that support the visual representation of the body as a whole [26]. Beyond visual information, body schema refers to the dynamic, sensorimotor representation that allows feeling the own body in space during movements and to map other people’s actions into own internal states. This process seems to have important implications for the understanding of others’ sensorial, motor, emotional and cognitive states through embodiment [27,28,29]. Sensorimotor body representations are also involved in simulating actions and movements, which have been widely investigated through paradigms of motor imagery, ref. [30] namely the ability to mentally simulate movements of body-parts [31,32] or whole-body stimuli [33]. Motor imagery is a distinct, dissociable process compared to visual imagery, which is engaged by mental simulation of non-bodily stimuli [30,34]. Indeed, they follow diverse developmental trajectories [35] and are underpinned by different neurocognitive networks [36], involving both temporo-parietal [37,38] and fronto-parietal areas, with an overlap for these latter areas on circuits associated with action observation [39]. The network subtending motor imagery also engages subcortical and cerebellar regions [39].

With respect to body processing in developmental age, configural and holistic processing have been documented in typically developing children aged 6–11 years, with no differences between younger and older children [40]. This suggests that the use of refined perceptual strategies to process body stimuli could appear early in life, in keeping with infant research on face perception [41,42]. Further, as early as 6 years old children show adult-like configural and holistic processing of body stimuli, suggesting that higher-order cognitive abilities, such as attention and executive functions, do not directly affect body visual perception [40]. In contrast, corporeal self-recognition has been identified already starting from 4 years of age in typically developing children [43] and the development of the body schema has been suggested to change significantly in children from 6 to 11 years of age [35].

In keeping with the existence of specialized networks for body representation [44,45], neuropsychological studies have largely documented specific disorders in either visual [40,46,47,48,49] or sensorimotor aspects [50,51] of body processing after acquired brain damage in adulthood. Less is known about the neuropsychology of body processing in developmental age. Nevertheless, exploring this topic seems to be crucial to understand how brain damage in developmental age may alter the normal process of developing body representations, also considering that data on adult patients may not be completely generalizable to children [25,52,53]. Indeed, hemispheric differences in processing self and “others” body part have been described [43]. For both children and adults, the processing of self and others’ body parts has been found to rely on different networks, which can be selectively impaired after a specific hemispheric lesion. Indeed, left and right brain damage differently affect the processing self and others’ body parts [43]. This functional independence is thought to correspond to anatomical independence [43].

Previous developmental neuropsychology studies have mainly investigated body representation deficits in children and adolescents with cerebral palsy (who typically show relevant motor deficits) [12,54,55,56], documenting impairments in both holistic body perception [54] and motor imagery [12,55,57]. Also a study conducted on survivors from a pediatric brain tumor [58] found a disruption in holistic body processing. The presence of alterations in holistic body processing but not in configural body processing in patients with both cerebral palsy [54] and infratentorial tumors [58] may suggest that altered sensorimotor experience of one’s own body due to motor deficits may negatively affect the development of holistic processing of others’ bodies in late childhood. The study revealed that patients with infratentorial tumors, which usually lead to significant motor impairments, showed alterations in mental imagery process, while patients with supratentorial tumors, which affect the brain cortex, presented more visual processing deficits. The presence of alterations in holistic body processing but not in configural body processing both in patients with cerebral palsy and infratentorial tumors may suggest that altered sensorimotor experience of one’s own body due to motor deficits may negatively affect the development of holistic processing of others’ bodies in late childhood [54,58].

The present study is aimed at investigating the performance related to body processing in children and adolescents with traumatic brain injury (TBI), with the objective to extend knowledge on the topic by considering a pediatric neurological population whose body representation disorders have been scantly tested before. Compared to cerebral palsy, a damage related to TBI occurs in a brain that has previously followed typical developmental trajectories and usually implies motor deficits to a lower extent. At the same time, TBI is different from the acquired damage associated with brain tumor, as it is not a progressive disease and does not require adjuvant therapies during the recovery course (i.e., radiotherapy or chemotherapy), which are known to progressively alter cognitive functioning [59]. It is important to note that the experimental paradigms used in the present study are the same of the ones adopted in previous studies related to pediatric populations, focusing on cerebral palsy [54], brain tumor [58] and preterm birth [60]. This fact ensures the comparability of data across studies, allowing more in depth reflections on the development of body processing in neurological populations. Specifically, in order to examine visual body representation, children and adolescents were proposed with a visual body recognition task relying on configural and holistic processing [15,19,61,62]. The ability of processing the body schema was investigated through a visuospatial imagery paradigm assessing mental rotation of whole body vs. non-bodily stimuli (i.e., letters) requiring, respectively, first-person mental transformations and object-based transformations [35]. The performance of TBI children in visual body recognition and visuospatial imagery tasks was compared to that of age-matched typically developing (TD) children and adolescents in order to detect anomalies in competences of TBI patients as compared to age-matched typical development.

Due to the different types of neural alterations associated with TBI, children included in this study were split into two groups on the basis of neuroradiological records, those with grey matter damage (GMD) and those with diffuse axonal injury (DAI). Drawing from previous research examining body processing [54,58], we expected that children and adolescents with TBI could be less efficient than TD children in both tasks. In detail, for visual body recognition, we expected that TD children showed both configural and holistic processing, while TBI children could be impaired particularly in using more refined holistic processing strategies (as reflected in the composite illusion effect), in keeping with findings on children with cerebral palsy and brain tumor [54,58]. Furthermore, it could be anticipated that children with DAI could present more difficulties of visual body processing, on the basis of data from previous literature suggesting the central role of white matter integrity for the visual processing of social stimuli in adult populations [63,64,65]. Finally, also for the visuospatial imagery task, a worse performance was expected for the DAI than the GMD subgroup due to the higher involvement of cortico-subcortical circuits in this cognitive process [66,67,68], which could lead to hypothesize that individuals with significant axonal damage could face difficulties in integrating motor and visual information.

## 2. Materials and Methods

### 2.1. Participants and Inclusion Criteria

Participants were 33 children and adolescents aged 8–18 years with a diagnosis of TBI in post-acute or chronic phase, referred to the Severe Acquired Brain Injury Department of the Scientific Institute IRCCS E. Medea (Bosisio Parini, Italy) for clinical and functional evaluation and rehabilitation interventions. TBI followed road traffic and domestic accidents; none had electrical injury [69]. Inclusion criteria were: (i) absence of severe sensory or motor deficits that could prevent task execution; (ii) proper comprehension and speaking of the Italian language; (iii) the Full Scale Intellectual Quotient (FSIQ) of the age corresponding Wechsler Intelligence Scale (Wechsler Intelligence Scale for Children Fourth Edition-WISC-IV or Wechsler Adult Intelligence Scale Fourth Edition-WAIS-IV) being >55, and at least one of the intellectual indices (i.e., verbal comprehension index-VCI- or perceptual reasoning index -PRI-) being ≥70, to exclude a global intellectual delay. The WISC-IV [70,71] and the WAIS-IV [71,72] have demonstrated to have good validity and reliability parameters in the Italian population, above all when considering the measurement of global intelligence. Specifically, the average reliability of WISC-IV is 0.96, comparable with data of the English version [71,73]. The revision of the data of the WISC-IV Italian test manual highlighted that also internal consistencies, interrater agreement, test–retest stability, and standard errors of measurement are in line with those of the English version [73]. With respect to WAIS-IV, the average reliability (0.97) and the four-factor structure are comparable with data of the English version [71,74]. This way, we ensured that all participants could understand and execute the tasks. Eligible participants were identified by the attending physician after a review of medical records. Their parents were approached by a research assistant who provided full information about the study and, in case of assent to partake, asked them to sign a written informed consent. In case of subjects being of legal age, the informed consent was signed directly by them. The study was approved by the Ethical Committee of the Scientific Institute IRCCS E. Medea (Prot. n. 024/15-CE) and was conducted in accordance with recommendations from the 1964 Declaration of Helsinki. For each participant, the following demographic and clinical measures were collected: gender, age at the event, age at evaluation, time since the event, hand laterality as assessed through a standard handedness inventory [75], WISC-IV intellectual indices (FSIQ; Verbal Comprehension Index-VCI; Perceptual Reasoning Index-PRI) and detailed lesion site and classification (i.e., grey matter damage -GMD; diffuse axonal injury -DAI) which was obtained by reviewing magnetic resonance imaging reports. Specifically, on the basis of neuroradiological records, children were split into two groups: a group with grey matter damage (GMD), which affects specific or multiple grey matter locations in absence of relevant damage to connective fibers, and a group with diffuse axonal injury (DAI), which primarily affects white matter and tissues at the intersections between grey and white matter [76]. TBI children were compared with two convenient samples of children and adolescents with typical development (TD) recruited from local schools and matched for gender and age: a group (N = 33) for the Visual Body Recognition Paradigm and a group (N = 31) for the Visuospatial Imagery Paradigm. TD participants and their parents were provided with all information about the study and parents were required to sign a written informed consent.

### 2.2. Procedure

Participants were placed in a silent room and they were administered by a researcher of the team the following experimental paradigms: (i) the Visual Body Recognition Paradigm, assessing the inversion effect (configural processing) and the composite illusion effect (holistic processing); this paradigm has been tested in a large sample of both adult and young subjects [40], showing reliability in detecting the body inversion and composite illusion effects in children as young as 6–7 years old; (ii) the Visuospatial Imaginary Paradigm, assessing motor imagery and mental imagery with bodily and letter stimuli, respectively. Both the paradigms have already been administered to healthy and clinical pediatric populations [54,58,60], demonstrating to be feasible in children with brain damage. No evidence of practice effect across the tasks was reported [54,58,60], since the two paradigms rely on different cognitive processes. To support the children and adolescents during the tasks, they were allowed to have a break for a minute between the different blocks of each paradigm and, if they asked, instructions on the task were repeated to them before each block. No feedback on accuracy was provided. To administer the experimental paradigms to participants, a PC laptop connected to 15.4 inch LCD monitor (resolution, 1024 × 768 pixels; refresh frequency, 60 Hz) was used. Participants were placed at a distance of approximately 60 cm from the computer monitor and responded using the left or right button of the mouse. The order of task administration was counterbalanced across participants. The experiment was run by using the E-prime 2 software package (Psychology Software Tools, Pittsburgh, PA, USA). A detailed description of each experimental task is provided below and the experimental procedure is depicted in Figure 1.

In each trial of Visual Body Recognition Paradigm, participants were asked to indicate whether two figures were the same or different with respects to the upper side. In the Visuospatial Imagery Paradigm participants were presented a single image (of a male or a female individual or of letter F) on each trial and had to judge whether the gray hand for the bodily image and the gray square for the letter F were at the left or at the right of the stimulus.

#### 2.2.1. Visual Body Recognition Paradigm (Configural and Holistic Processing)

Participants were required to perform delayed same-different judgments on color pictures of body postures. Stimuli were pictures of two boys and two girls aged 8 years displaying 6 different body postures, with various displacements of lower and upper limbs. The postures had no emotional content or symbolic meaning. Children depicted in the pictures were wearing the same grey/blue or pink/yellow t-shirts and shorts and were photographed while assuming the same set of body postures. The pictures were taken from frontal or sideway perspective and were displayed on a white background, subtending a 539 × 737 pixel area. For each of the 24 original pictures, a paired stimulus was created by combining the upper half of the body with the lower half of the body picture of a different model assuming the same posture and matched for gender. The Adobe Photoshop software (Adobe Systems Inc., San Jose, CA, USA) was used to digitally edit pictures. Twenty-four pairs were obtained, with the two stimuli in each pair having the same upper half but a different lower half. In front of a sequence of two body stimuli, participants were required to detect whether the upper part of the second stimulus was the same or different as compared to the upper part of the first stimulus. In the same-response trials, each stimulus was presented with the matching stimulus of the pair that had the same upper half but a different lower half. In contrast, in the different-response trials, each stimulus was presented with a stimulus of a different pair created from the same models and having different upper and lower parts. To evaluate the inversion and the composite illusion effects, stimuli were presented, respectively, upright or inverted (orientation condition) and aligned or misaligned (alignment condition). For the orientation manipulation, inverted stimuli were rotated at 180° along the horizontal axis, being reversed upside down. For the alignment manipulation, misaligned stimuli were obtained by shifting the lower body part to the right along the horizontal axis, starting at the middle of the upper body half. No gap was left between lower and upper body parts, in accordance with the procedure of previous studies reporting reliable composite illusion effect for bodies in both adults [77] and children and adolescents [43]. The children’s face was maintained but was scrambled: this choice was made to avoid interferences on inversion effects generated by headless bodies (see 105, 106), but at the same time to prevent face identity discrimination [78].

In line with previous literature [79], a partial design with a 2:1 proportion of same vs. different response trials was used. A total of 144 trials were administered, divided into 6 different blocks, each one consisting of 16 same- and 8 different-response trials. Before starting the experiment, oral and written instructions on the task were provided to participants. In order to verify comprehension of task rules and methods, they were presented with 8 practice trials, which were not considered for statistical analyses. Each trial started with the presentation of a central fixation cross lasting 1000 ms. Subsequently, the first stimulus was presented for 1500 ms, followed by a random-dot mask (76° × 76° in size; duration between 550 and 690 ms) obtained by scrambling body stimuli. The probe stimulus appeared immediately after the disappearance of the random-dot mask and remained on the screen until a response was given or for a maximum of 3500 ms. Participants had a maximum interval of 5000 ms from the onset of the probe stimulus to respond. In each trial, the paired stimuli had the same orientation and alignment but had different lower parts, while the upper parts could be either the same or different. Participants had to respond as quickly and accurate as possible by pressing the left or the right button on the computer mouse, corresponding, respectively, to a same or a different response. The subsequent trial appeared after an interval of 2500 ms.

#### 2.2.2. Visuospatial Imagery Paradigm (Motor and Visual Imagery)

The Visuospatial Imagery Paradigm required participants to perform right-left judgments on body stimuli (drawings representing a female-like or a male-like body manikin), and non-social stimuli (two differently written “F” letters, with the lower arm having the same or a smaller length than the upper arm). These two types of stimuli were presented in separate blocks, and participants were asked to perform a laterality judgement task. In particular, the body drawings had one hand marked in grey and participants were asked to judge whether the marked hand was the right or the left one, according to the manikin’s perspective. The body drawing stimuli could be shown in a front view or in a back view condition. In this latter condition, the stimuli were presented according to the child’s perspective, so that no mental transformation was required. In contrast, front-view stimuli required a first-to third-person perspective transformation to be responded (first-person transformation). The letter F was presented with a grey square on one side and participants were asked to judge if the square was on the left or the right side of such a stimulus. The letter F could be presented in the canonical position (unturned condition) or rotated at 180° around its vertical axis (turned condition). In the turned condition participants had to operate a mental transformation of the object to respond (object-based transformation). Thus, both the front view condition in the body task and the turned condition in the letter task required a mental transformation and were, consequently, expected to lead to increased response times and/or error rates as compared, respectively, to the back-viewing bodies and unturned letters [35]. Body and letter stimuli had the same dimensions along the vertical and the horizontal axes (600 × 600 pixels). They were presented at the center of the screen until a response was given. A 1 s interval was allowed between trials. The body and letter stimuli were presented in separate 64-trial blocks, of which 32 required a mental transformation (front-viewing bodies or turned letters) and 32 did not (back-viewing bodies; unturned letters). Trial order was randomly defined. Similarly, manikin gender, letter F type and left/right response trials were randomly presented within each block. Judgements on body and letter transformations were matched in terms of complexity and axis of mental transformation [35]. Participants were asked to respond to each trial as fast and accurately as possible, by pressing the right or the left button of the mouse, which corresponded, respectively, to a right or left judgement response.

### 2.3. Data Handling and Statistical Analyses

Participants with TBI were split into 2 groups according to the principal damage to either white or grey matter (i.e., DAI—vs. GMD). Preliminarily, Student’s *t*-tests and Chi^2^ tests were adopted in order to control for differences in demographic and clinical variables between the TBI group and the TD children group and among the 2 clinical groups.

For the Visual Body Recognition task, in line with previous literature on the composite illusion effect [62,79], only the same-condition trials were analyzed. For both the tasks, we excluded trials with anticipated or out-of-time responses (RT < 150 ms or > 5000 ms). With the aim to consider possible speed-accuracy trade-off effects, we computed the Inverse Efficiency (IE) index as the ratio between Reaction Times (RTs) and Accuracy, so that lower IE values corresponded to better task performance while higher values indicated worse performance. For the Visuospatial Imagery Paradigm, it is to note that 2 TBI participants (one with GMD, the other with DAI) were not able to perform the mental rotation when the body was presented in front-view (Accuracy < 50%) and were thus excluded from further analyses, as well as 2 matched control-subjects (N = 31 per group) For each task, the IEs were entered into a 3-way mixed-model, repeated-measures 2 × 2 × 2 ANOVA with Group (i.e., TD vs. TBI) as between-subject factor. The within-subjects variables were Alignment (Aligned vs. Non-aligned) and Orientation (Upright vs. Inverted) or Stimuli (Body vs. Letter) and Transformation (Non-required vs. Required), as regards, respectively, the Visual Body Recognition task and the Visuospatial Imagery Paradigm. For the Visual Body Recognition task, a planned follow-up ANOVA was conducted within the TD children group to further investigate the presence of body inversion and composite illusion effects. Conversely, the clinical group (GMD vs. DAI) was entered as between-subject factor into a 3-way mixed-model, repeated-measures ANOVA with the same repeated-measure variables as above.

The group sample size was determined a priori through a power analysis with the G*Power software [80]. On the basis of a previous study assessing visual body perception in children and adolescents with brain tumor compared to healthy peers [58] (η^2^_p_ = 0.123) and using the “as in SPSS” option, an expected effect size of f (U) = 0.375 was estimated. Thus, considering a 3-way mixed-model, repeated-measures 2 × 2 × 2 ANOVA (numerator df = 2) and setting the significance level at 0.05, and the desired power (1 β) at 0.80, we obtained a total sample size of 60 participants (30 per group). As a 10% drop-out rate was expected, we implemented an oversampling of 33 individuals per group. The significance threshold was set at *p* < 0.05 for all statistical tests. Multiple-way interactions were analyzed adopting Duncan’s post-hoc test correction for multiple comparisons. Effect sizes were reported as partial Eta squared (η^2^_p_), adopting conventional cut-offs of η^2^_p_ = 0.01, 0.06; and 0.14 for small, medium, and large effect sizes, respectively [81]. Data were reported as mean and standard error of the mean (SEM) unless otherwise stated. Analyses were performed by means of the Statistica software version 7 (Statsoft, Tulsa, OK, USA).

## 3. Results

### 3.1. Demographic and Clinical Variables

Preliminary analyses confirmed that the TBI group (mean age: 13.48 years; gender: 21 males) did not differ for age and gender from the TD children group recruited for the Visual Body Recognition task (mean age: 12.24 years; gender: 20 males; t_64_ = 1.56, *p* = 0.123; Chi^2^ = 0.06, *p* = 0.80) or for the Visuospatial Imagery Paradigm (mean age: 12.58 years; gender: 20 males; t_60_ = 0.89, *p* = 0.379; Chi^2^ = 0.07, *p* = 0.79).

In the TBI group, 16 participants had a GMD (13.38 years, 7 females) and 17 had a DAI (13.59 years, 5 females). Results did not highlight differences for demographic and clinical variables (all *p* > 0.076), even after the exclusion of 2 subjects in the Visuospatial Imagery Paradigm (all *p* > 0.140). Demographic and clinical variables of the clinical groups are reported in Table 1.

### 3.2. Visual Body Recognition Task

This task could be performed by all 33 subjects with TBI, which were compared with 33 TD children. Raw data for Accuracy and RTs for the Visual Body Recognition task are reported in Table 2.

The three-way (2 × 2 × 2) mixed model repeated measures analysis highlighted non-significant effects (all F < 3.30, all *p* > 0.073) but a significant Alignment × Group interaction effect (F_1,62_ = 7.43, *p* = 0.008, η^2^_p_ = 0.10), pointing to a diverse presence of the body composite illusion effect in the 2 groups. Duncan post-hoc tests revealed that TD children performed significantly better with non-aligned stimuli (1306.74 ± 131.85) than with aligned stimuli (1386.76 ± 139.55; *p* = 0.002), according to the use of holistic processing for body stimuli, while the misalignment did not facilitate task execution in the TBI group (aligned: 1272.67 ± 139.55, non-aligned: 1288.68 ± 131.85; *p* = 0.523). For the TD group, the planned follow-up ANOVA confirmed a significant main effect of Alignment (F_1,32_ = 9.76, *p* = 0.004, η^2^_p_ = 0.23) and revealed a significant Alignment × Orientation interaction (F_1,32_ = 4.62, *p* = 0.039, η^2^_p_ = 0.13), thus pointing to the use of both configural and holistic processing for body stimuli. In line with the body inversion effect, TD children showed better performance for upright compared to inverted body stimuli when presented aligned (1244.26 ± 73.33 vs. 1529.26 ± 278.21; *p* < 0.001) as well as misaligned (1128.14 ± 49.13 vs. 1485.33 ± 272.03; *p* < 0.001). Moreover, they performed better with upright misaligned stimuli than with upright aligned bodies (*p* < 0.001), as expected by the body composite illusion, and compared to the other conditions (all *p* < 0.001). Notably, when bodies were inverted, misalignment did not facilitate the detection of differences (*p* = 0.074), since configural processing interfered with the higher-level holistic processing, partially counteracting the expected advantage for misaligned stimuli (Figure 2).

The follow-up ANOVA comparing the two clinical subgroups revealed a marginally significant effect of group (F_1,31_ = 4.11, *p* = 0.051, η^2^_p_ = 0.12), with DAI participants showing a worse performance than the group with GMD in the task. However, all other effects were non-significant (all F < 0.70, all *p* > 0.408), thus confirming the absence of both holistic and configural processing for body stimuli in both clinical groups (Figure 3).

### 3.3. Visuospatial Imagery Paradigm

Raw data for Accuracy and RT are reported in Table 3.

This task was appropriately performed by 31 subjects with TBI, which were compared with 31 TD children. The three-way (2 × 2 × 2) mixed model repeated measures ANOVA revealed significant Group (F_1,60_ = 15.58, *p* < 0.001, η^2^_p_ = 0.21) and Transformation (F_1,60_ = 18.55, *p* < 0.001, η^2^_p_ = 0.24) main effects, better qualified by their significant interaction (F_1,60_ = 7.56, *p* = 0.008, η^2^_p_ = 0.11). Post-hoc tests highlighted that the TBI group showed a worse performance for stimuli requiring mental rotation than for stimuli presented in congruent positions (3958.75 ± 522.65 vs. 2097.18 ± 331.72; *p* < 0.001), while such a difference was not reliable in the TD group (1601.76 ± 522.65 vs. 1190.66 ± 331.72; *p* = 0.275). Between-group comparisons did not indicate a significant difference for stimuli not requiring mental rotation (*p* = 0.083) but pointed to a specific impairment of the TBI group compared to TD children when they had to operate mental transformations (*p* < 0.001). The main effect of Stimulus and the interactions involving it were all non-significant (all F < 0.91, all *p* > 0.342) (Figure 4).

The ANOVA comparing the 2 clinical subgroups confirmed a significant effect of Transformation (F_1,29_ = 12.42, *p* = 0.001, η^2^_p_ = 0.30), with worse performance for rotated compared to congruent stimuli, while all the other effects were non-significant (all F < 1.06, all *p* > 0.311) (Figure 5). These results clarified that participants with TBI had a general difficulty in operating visuospatial mental transformation, independently from the type of brain injury and from the presented stimulus.

## 4. Discussion

In this study, we investigated whether children and adolescents with TBI may show difficulties in body representation, and which aspects of this complex ability are more deeply affected by damage, more involving the grey matter (group with GMD) or the white matter (group with DAI). Specifically, we administered a visual body recognition task, assessing the presence of configural and holistic processing, and a visuospatial imagery task, requiring participants to perform first- or third-person rotations on whole body or letter stimuli, respectively. The performance of patients with TBI was compared with that of children and adolescents with TD.

In relation to visual body representation, patients with TBI showed reduced body inversion and body composite effects as compared to TD children, pointing to deficits of both configural and holistic processes in the former population. Previous studies in children and adolescents with cerebral palsy [54] and brain tumor [58] documented impaired holistic processing, but spared configural body processing in these pediatric neurological populations. Both perceptual strategies are involved in the so-called body structural description, and allow a more efficient processing of bodies (and faces) than the detail-based processing adopted for objects [16]. However, they rely on different types of stimulus processing. Indeed, configural processing relies on the detection of first-order (e.g., the head is above the shoulders) and second-order relations (e.g., the specific width of the trunk) between different parts [19,61]. Conversely, holistic processing consists in matching the presented stimulus with a category-specific template and is considered a more refined processing strategy. Overall, findings on children and adolescents with TBI indicated that a brain damage occurring during development may alter the evolution trajectory of core visual mechanisms underlying visual body processing, which are found to be present even at 6–7 years of age and to be independent from the evolution of other cognitive abilities [43]. This indicates the great impact of a brain lesion on the development of refined perceptual strategies devoted to the analysis of specie-specific stimuli such as bodies, possibly hindering the social advantages associated with such an ability [82,83]. It has been previously shown that different type of brain damage may differently alter body representation and its relationship with cognitive and emotional disturbances [84] Accordingly, deficits in the processing of others’ bodies have been found in pathologies that frequently include social difficulties, such as anorexia nervosa and bulimia [78], schizophrenia [85,86] and autism spectrum disorder [87,88], supporting this link.

When considering the specific nature of the TBI, that is DAI or GMD, a marginally significant difference in the ability to perform visual body processing was found, with children with DAI showing worse performance. Even though for both the clinical subgroups the absence of the configural and holistic processing was confirmed, patients with DAI demonstrated to be less rapid in providing body discrimination responses. No differences in visual and motor abilities between the two clinical subgroups were found, as well as no differences were observed in general cognitive abilities, including either verbal or perceptual domains. This might lead to hypothesize that the worse performance of patients with DAI could be associated to the specific features of their neural damage, which involve more disruption of fiber connections as compared to GMD [76]. In line with this hypothesis, body image disturbances in women with anorexia nervosa have been associated to a reduced connectivity between ventral occipito-temporal cortices specifically involved in body processing [89,90]. Previous research also suggested that damage to the connectivity between parieto-occipital regions and right temporal areas may be related with more severe impairments in the perception and understanding of others’ actions [91]. In a similar vein, the deficits of patients with developmental prosopagnosia, who show impaired face and body visual recognition [92], have been associated with disruption of connectivity within ventral occipito-temporal areas. These findings highlight the central role of white matter integrity for supporting the processing of social stimuli [63,64,65]. However, the specific effects of DAI vs. GMD to specific neuroanatomical structures on configural and holistic processing of body stimuli need to be investigated in further studies with larger sample, also considering the important clinical implications that this could have with respect to the emotional and relational abilities of individuals with neurological conditions [84].

With respect to mental imagery, we found that children and adolescents with TBI showed significantly lower abilities than TD peers in performing mental transformations with both body and object stimuli, thus displaying deficits in both motor and visual imagery abilities. This keeps with previous studies showing motor imagery abilities in children with TBI [93]. Motor imagery is a distinct, dissociable process compared to visual imagery engaged by mental simulation of non-bodily stimuli [30,34], as suggested by the fact that they follow diverse developmental trajectories [33] and are underpinned by different neurocognitive networks [36]. Considering this aspect, our finding seems to indicate that the impairment of children with TBI was not limited to the areas of body processing but more generally ascribable to visual-spatial circuits [39]. Thus, such a visuo-spatial deficit may negatively impact not only orientation skills and academic abilities requiring the processing of the spatial information, but also the sensorimotor representations of the self and others.

Based on the above considerations, it would be useful to provide patients with TBI with ad hoc rehabilitation aimed at boosting both visual-spatial abilities and specific competences associated with body processing. Previous studies found that exposure-based cognitive-behavioral therapy could be an effective treatment for visual body perception [91,94,95], while rehabilitation approaches addressing body-related cognitive process in addition to physical exercise (motor activation of upper and lower limbs or extremities, if feasible) could be useful for mental imagery abilities [96,97]. The rehabilitation of TBI patients could also benefit from the simulation of body movements or motor imagery activities within virtual reality environments, which could allow overcoming the motor difficulties in real world interactions and provide a precise control of stimuli and results, thus favoring the collection of evidence-based data on treatment effectiveness [98,99,100,101,102].

The limitations of this study should be acknowledged. First, even though the sample size of the study was established by performing an a priori power analysis, the relatively small number of enrolled patients may limit the generalizability of findings. Thus, future studies are needed to confirm and refine our findings. Generalization may also be affected by biased selection of participants with respect to the whole clinical population of children and adolescents with TBI, due to inclusion of only those subjects not presenting severe motor or sensory deficits, which would have undermined the possibility to perform the study tasks. Second, issues related to the methodology of the proposed tasks to assess body processes have been reported by previous research. In particular, as regards the visual body recognition task, criticism has been moved on the possible interference that both inversion [15] and composite illusion effects [103] could have on configural and holistic processing. It has also been suggested to use a complete design rather than a partial one for the composite illusion task [104,105]. However, previous research on holistic processing in pediatric patients have usually adopted the latter method [106], thus supporting this methodological choice for the present study. With respect to the visuospatial imagery task, the rotated perspective condition of bodies (representing stimuli in the front view) has been criticized in its ability to activate own-body mental rotation [107]. Indeed, it has been suggested that children could instead answer by inverting the left-right and front-back axis with no rotation of their mental position. Despite this possibility, previous studies conducted on pediatric populations [35] found that this visuo-spatial imagery paradigm is reliable to detect the dissociation between object- and viewer-transformation ability [58,108].

In conclusion, children and adolescents with TBI exhibited altered body processing, both in the visual elaboration of body stimuli (configural and holistic processes) and in the ability to mentally transform body images by performing mental rotations, in line with general visual-spatial deficits in mental transformations that also affect visual imagery of external objects. Such a finding highlights the importance of the rehabilitation of general visuo-spatial abilities after a TBI in order to boost not only spatial orientation and academic achievements, but also for the adequate development of social and embodied cognition. At the same time, it appears that a need to offer specific rehabilitation on body processing, such as exposure-based cognitive-behavioral therapy to improve visual body perception [90,94,95] and motor imagery and action observation to train imagery abilities [96,97]. Further, in order to provide detailed information on the neural bases underlying body processing disorders in children and adolescents with TBI, future research should use detailed neuroanatomical mapping to investigate which specific circuit damage is associated with more severe body processing impairments.

## Figures and Tables

**Figure 1 brainsci-12-00962-f001:**
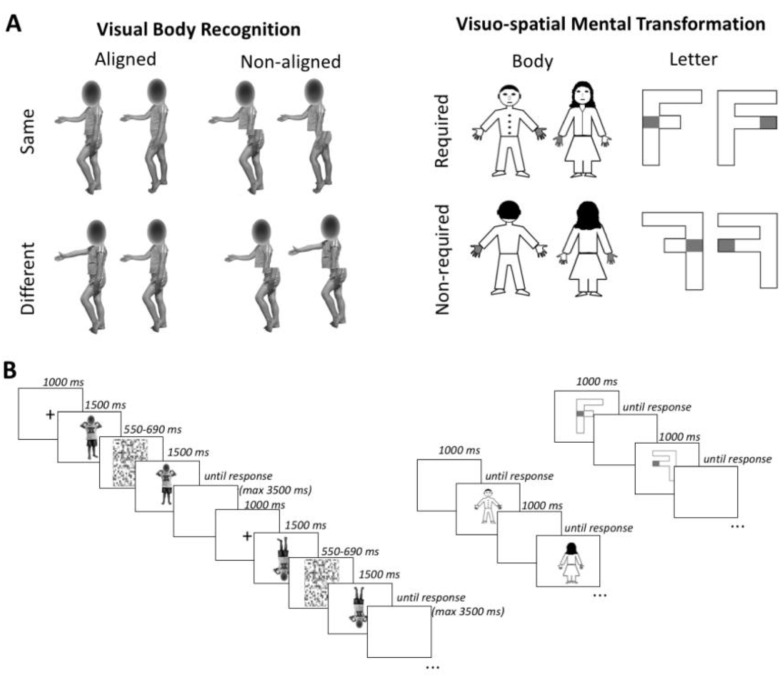
Experimental Paradigms. (**A**)—Illustration of stimuli of the Visual Body Recognition Paradigm (**left side**) and the Visuo-spatial Imagery Paradigm (**right side**). (**B**)—Schematic representation of the time line of the trials.

**Figure 2 brainsci-12-00962-f002:**
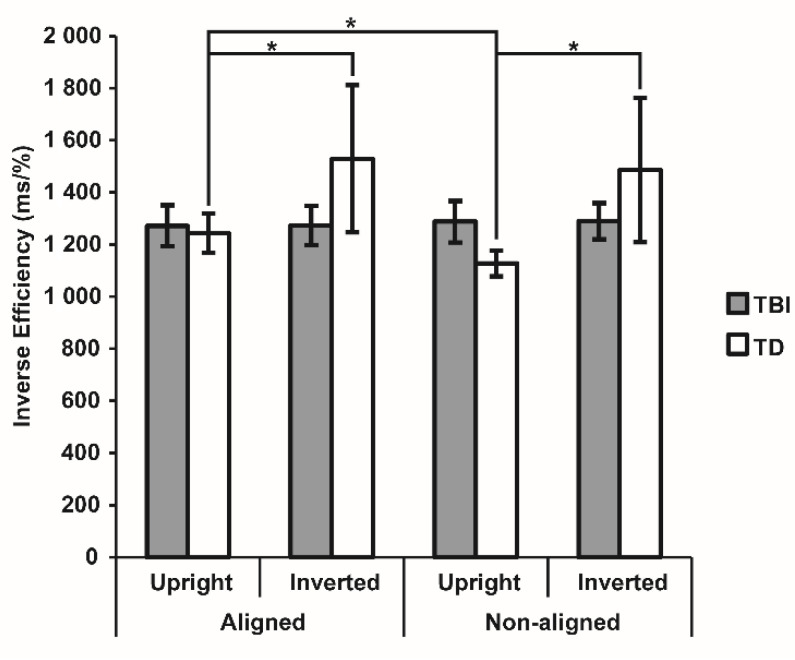
Inverse effect (IE) for each condition of the Visual Body Recognition task for the two groups. Bars indicate Standard Error of the Mean of measurements of 16 same-response trials in six blocks (N = 96) for 33 children with traumatic brain injury (TBI) and 33 TD (typically developing) children. Dotted black lines show within-group comparisons, with asterisks indicating significant *p* < 0.05.

**Figure 3 brainsci-12-00962-f003:**
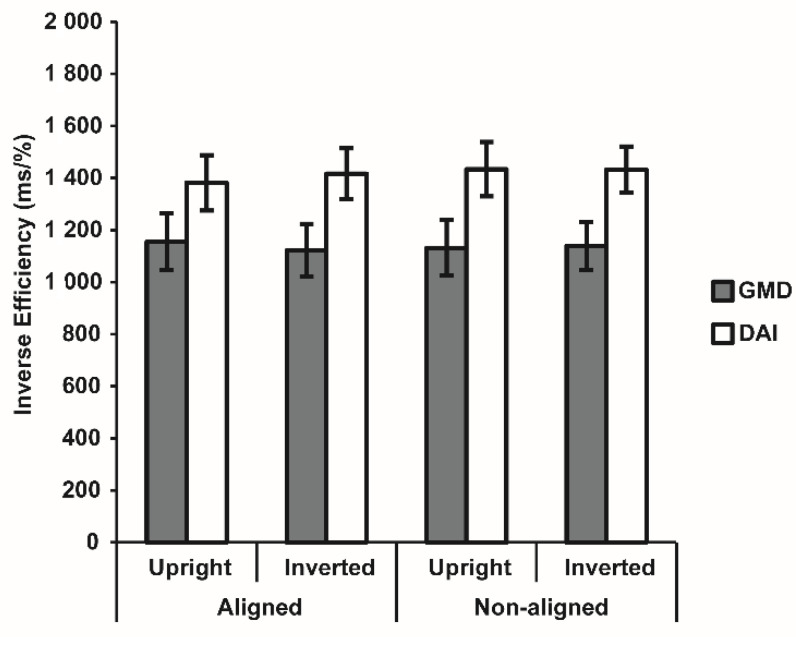
Inverse effect (IE) for each condition of the Visual Body Recognition task for the two clinical groups. Note. Bars indicate Standard Error of the Mean of measurements of 16 same-response trials in six blocks (N = 96) for 16 participants with grey matter damage (GMD) and 17 participants with diffuse axonal injury (DAI).

**Figure 4 brainsci-12-00962-f004:**
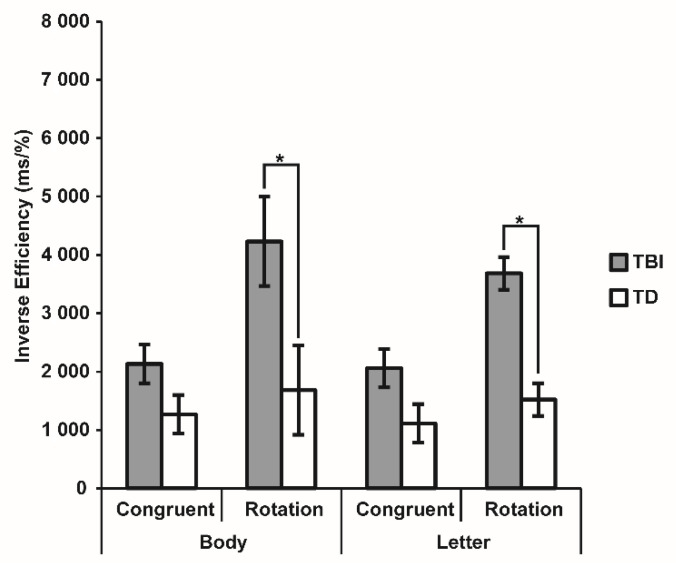
Inverse Effect (IE) for each condition of the Visuospatial Imagery Paradigm for the two groups. Bars indicate Standard Error of the Mean of measurements in 64 Body and 64 Letter trials for 31 TBI and 31 TD children. Dotted black lines show between-groups comparisons, with asterisks indicating significant *p* < 0.05.

**Figure 5 brainsci-12-00962-f005:**
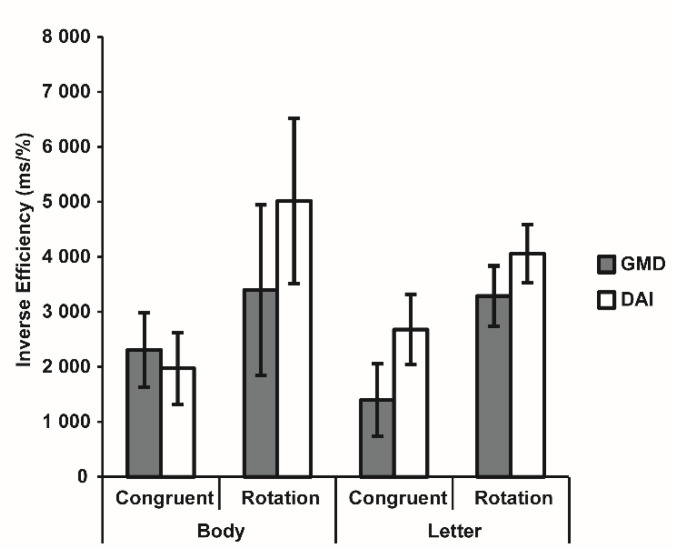
Inverse Effect (IE) for each condition of the Visuospatial Imagery Paradigm for the two clinical groups. Bars indicate the Standard Error of the Mean of measurements in 64 Body and 64 Letter trials for 15 participants with GMD and 16 participants with DAI.

**Table 1 brainsci-12-00962-t001:** Demographic and clinical variables of patients with grey matter damage (GMD) and diffuse axonal injury (DAI).

	GMD (N = 16)		DAI (N = 17)		t/Chi2	*p* Value
	Mean (SD)	N (%)	Mean (SD)	N (%)		
Demographic variables						
Sex (males)		9 (56%)		12 (71%)	0.73	0.392
Age at evaluation (years)	13.38 (2.58)		13.59 (2.18)		0.26	0.799
Clinical variables						
Time since TBI (years)	5.93 (4.66)		3.19 (3.89)		1.84	0.076
GCS	7.69 (3.02)		6.88 (3.67)		0.68	0.499
Motor impairments		9 (56%)		11 (65%)	0.25	0.619
Visual impairments		10 (63%)		6 (35%)	2.44	0.118
Cognitive functioning						
FSIQ	98.25 (16.21)		86.17 (21.47)		1.81	0.079
VCI	101.38 (165.60)		94.82 (16.52)		1.17	0.251
PRI	103.50 (18.68)		93.00 (20.64)		1.53	0.136

DAI = Diffuse Axonal Injury; FSIQ = Full Scale Intellectual Quotient; GCS = Glasgow Coma Scale; GMD = Grey Matter Damage; PRI = Perceptual Reasoning Index; SD = Standard Deviation; TBI = Traumatic Brain Injury; VCI = Verbal Comprehension Index.

**Table 2 brainsci-12-00962-t002:** Accuracy and Reaction Time in each experimental condition for the two groups (TBI group and TD group).

		Accuracy (%)		RT (ms)	
Alignment	Orientation	TBI Group	TD Group	TBI Group	TD Group
Aligned	Upright	86.73 ± 1.63	87.85 ± 1.36	1076.59 ± 54.40	1083.12 ± 62.44
	Inverted	87.53 ± 1.62	85.79 ± 2.77	1093.73 ± 54.35	1114.89 ± 73.58
Non-aligned	Upright	86.30 ± 1.80	90.48 ± 1.19	1082.82 ± 55.48	1022.27 ± 50.47
	Inverted	86.45 ± 1.59	86.15 ± 2.64	1095.07 ± 50.54	1086.08 ± 67.86

RT = reaction time; TBI = traumatic brain injury; TD = typically developing. Percentages (%) of accuracy refer to correct responses.

**Table 3 brainsci-12-00962-t003:** Accuracy and Reaction Time in each condition of the Visuospatial Imagery Paradigm for the two groups.

		Accuracy (%)		RT (ms)	
Stimulus	Transformation	TBI Group	TD Group	TBI Group	TD Group
Body	Congruent	84.39 ± 4.21	94.84 ± 0.80	1256.20 ± 89.58	1195.26 ± 76.83
	Rotation	68.71 ± 6.07	91.26 ± 1.29	1459.24 ± 1117.63	1490.25 ± 108.96
Letter	Congruent	84.94 ± 3.56	96.00 ± 0.83	1380.92 ± 108.96	1066.58 ± 48.86
	Rotation	47.87 ± 1.62	90.52 ± 2.18	1644.77 ± 148.40	1328.25 ± 73.66

Data are reported as mean ± SEM. RT = reaction time; TBI = traumatic brain injury; TD = typically developing.

## Data Availability

The data presented in this study are available on request from the corresponding author. The data are not publicly available due to privacy restrictions.

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
