# Peer review of "Body Processing in Children and Adolescents with Traumatic Brain Injury: An Exploratory Study"

_brainsci, 2022, doi:10.3390/brainsci12080962_

Round 1
Reviewer 1 Report
Body processing in children and adolescents with traumatic brain injury: an exploratory study
Overall, it presents a topic of interest, which brings information to the scientific field. Some adjustments should be made to improve the quality of the paper (more detailed explanations of various aspects of the theoretical background, method, and analysis...).
Abstract: it is in line with the requirements of the journal: “1) Background; 2) Methods; 3) Results; and 4) Conclusion”, but in the method a further explanation on the design is required.
Keywords: as the main topic is the recognition of the lack of research in children and adolescents with TBI these words should be considered as keywords.
Introduction: The study is pertinent, mentioning a topic of current interest. The theoretical foundation intended to support the importance of the topic, but some of the changes and improvements are recommended:
- Insert a new paragraph to start with “Previous studies…” (line 37)
- Change the order of presenting “holistic and configural visual perception strategies” (lines 38-39) to be according to the subsequent explanation
- In general, the explanation of the topic is interesting, but it is not an integrated knowledge of the subject matter, but three distinct parts: one, general knowledge about body processing (lines 33-62); another, a brief explanation of how this processing is altered in people with brain disorders (lines 63-67); and finally, a brief reference to what happens in childhood and adolescence (lines 67-79). As the main topic is childhood and adolescence, it makes no sense that the introduction does not devote more time to this topic and, above all, that it does not do so in an integrated way. It is recommended that the first two paragraphs be revised to incorporate information on what happens in people with brain disorders and, more specifically, in children and adolescents.
- The first part of the last paragraph (lines 103-107) has no sense in the introduction since they are explaining the method. Keep only the hypothesis based on the theoretical fundations.
Materials and Methods: The sample is small, which is understandable for the type of study presented, but not for such a wide age range (10 years). Both aspects should be noted as limitations to generalizability. The selection of scales is relevant and well explained, although it is recommended to include data on reliability and validity to make them more rigorous, as well as to include citations to the authors of the scales. A further explanation is required on the procedure regarding the measures that were needed to support the children while they were answering the tasks. Difficulties in the children's performance of the task and how they adapted should be considered. If they do not exist, they should also be reported (they could present it after lines 229 and 261).
Results: data presented are clear and relevant. The analyses presented are well presented and clearly relevant to the type of study being carried out. The results provided in the study are particularly interesting. It is recommended to reduce the size of the table, to adjust it to the rest of the text font size, and for simplicity, not to combine the values of M/SD, N/% and t/chi-square as they can lead to confusion and misreading.
Discussion and conclusion: The discussion is well constructed, but again, they need to focus more deeply on the target population.
Cites and References: The references are relevant, but they require a thorough updating, since the authors are referring to the lack of research on their subject and they do not do so on the basis of up-to-date knowledge, but rather 87.23% of their references are older than the last five years.
Author Response
Body processing in children and adolescents with traumatic brain injury: an exploratory study
Overall, it presents a topic of interest, which brings information to the scientific field. Some adjustments should be made to improve the quality of the paper (more detailed explanations of various aspects of the theoretical background, method, and analysis...).
Abstract: it is in line with the requirements of the journal: “1) Background; 2) Methods; 3) Results; and 4) Conclusion”, but in the method a further explanation on the design is required.
RE: thank you for this suggestion, we included more detailed information on the study design and procedure, in line with the limited word count of the Abstract.
Keywords: as the main topic is the recognition of the lack of research in children and adolescents with TBI these words should be considered as keywords.
RE: we added the indicated keywords.
Introduction: The study is pertinent, mentioning a topic of current interest. The theoretical foundation intended to support the importance of the topic, but some of the changes and improvements are recommended:
- Insert a new paragraph to start with “Previous studies…” (line 37)
RE: we inserted a new paragraph, as suggested.
- Change the order of presenting “holistic and configural visual perception strategies” (lines 38-39) to be according to the subsequent explanation
RE: we changed the order, as correctly suggested.
- In general, the explanation of the topic is interesting, but it is not an integrated knowledge of the subject matter, but three distinct parts: one, general knowledge about body processing (lines 33-62); another, a brief explanation of how this processing is altered in people with brain disorders (lines 63-67); and finally, a brief reference to what happens in childhood and adolescence (lines 67-79). As the main topic is childhood and adolescence, it makes no sense that the introduction does not devote more time to this topic and, above all, that it does not do so in an integrated way. It is recommended that the first two paragraphs be revised to incorporate information on what happens in people with brain disorders and, more specifically, in children and adolescents.
RE: thank you for this useful suggestion. We incorporated more information on body processing in children and adolescents and better described data on previous studies conducted in pediatric populations with brain disorder.
- The first part of the last paragraph (lines 103-107) has no sense in the introduction since they are explaining the method. Keep only the hypothesis based on the theoretical fundations.
RE: we moved this part to the Method, limiting the indicated sentence of the Introduction to the explication of the splitting of the sample into two groups, which is crucial to understand the hypotheses.
Materials and Methods: The sample is small, which is understandable for the type of study presented, but not for such a wide age range (10 years). Both aspects should be noted as limitations to generalizability. The selection of scales is relevant and well explained, although it is recommended to include data on reliability and validity to make them more rigorous, as well as to include citations to the authors of the scales. A further explanation is required on the procedure regarding the measures that were needed to support the children while they were answering the tasks. Difficulties in the children's performance of the task and how they adapted should be considered. If they do not exist, they should also be reported (they could present it after lines 229 and 261).
RE: we agree with the reviewer that the limited sample size might limit the generalizability of the findings and have declared this in the ‘Limitations’ section of the MS. Even considering the wide age range of the sample, our inclusion-exclusion criteria aimed to select those patients that could afford the administration of a relatively long computer testing session. However, the required sample size was calculated by using power analysis and was supported to be sufficient to test our hypotheses. The task administered have been taken from previous studies in pediatric population, but they have not reported psychometric properties. We reported, however, as required, the measures needed to support the children while they were answering the tasks.
Finally psychometric data of the scales WISC-IV and WAIS IV were reported.
Results: data presented are clear and relevant. The analyses presented are well presented and clearly relevant to the type of study being carried out. The results provided in the study are particularly interesting. It is recommended to reduce the size of the table, to adjust it to the rest of the text font size, and for simplicity, not to combine the values of M/SD, N/% and t/chi-square as they can lead to confusion and misreading.
RE: we modified Tables, as requested.
Discussion and conclusion: The discussion is well constructed, but again, they need to focus more deeply on the target population.
RE: we added more information on the target population in the discussion.
Cites and References: The references are relevant, but they require a thorough updating, since the authors are referring to the lack of research on their subject and they do not do so on the basis of up-to-date knowledge, but rather 87.23% of their references are older than the last five years.
RE: we thank the reviewer for prompting us to a refresh of the reference list. We have further searched for more recent papers on body processing in children and adolescents with TBI and have updated the reference list and introduction and methods section accordingly. Still, our claim of limited research on this aspect remains and the importance of conducting these studies was corroborated by evidence of correlation between altered body processing and cognitive and emotional processing (eg., Corallo et al. 2021).
Reviewer 2 Report
Corti et al. analyzed the performance related to body processing in children and adolescents with traumatic brain injury (TBI), with the objective to extend knowledge on the topic by considering a pediatric neurological population whose body representation disorders have never been tested before. It is interesting paper, authors used sufficient methods and there is not so much such analysis in current literature. However, the article require some clarifications. the language requires some improvement (especially citation format). What is more, authors should add information about the kind of acute brain injury. Where there any ie. electrical injuries? (Neurol Neurochir Pol 2021;55(1):12-23.). therefore, I suggest major revision.
Author Response
Corti et al. analyzed the performance related to body processing in children and adolescents with traumatic brain injury (TBI), with the objective to extend knowledge on the topic by considering a pediatric neurological population whose body representation disorders have never been tested before. It is interesting paper, authors used sufficient methods and there is not so much such analysis in current literature. However, the article require some clarifications. the language requires some improvement (especially citation format).
RE: thank you for your positive evaluation of our manuscript. We have double checked language use in the MS and have further screened uniformity in citation style use.
What is more, authors should add information about the kind of acute brain injury. Where there any ie. electrical injuries? (Neurol Neurochir Pol 2021;55(1):12-23.). therefore, I suggest major revision.
RE: as reported in the ‘Participants’ section, children had a previous diagnosis of traumatic brain injury (TBI) to be enrolled in the study, ad this was typically caused by road traffic or domestic accidents. None had electrical injuries. We have now specified this in the MS.
Round 2
Reviewer 2 Report
Authors respond to all my comments. I have no further remarks. I suggest acceptance of the paper.